# Versatile Cost Partitioning with Exact Sensitivity Analysis

**Primary Keywords:** *None*

## Abstract

Saturated post-hoc optimization is a powerful method for computing admissible heuristics for optimal classical planning. The approach solves a linear program (LP) for each state encountered during the search, which is computationally demanding. In this paper, we theoretically and empirically analyze to which extent we can reuse an LP solution of one state for another. We introduce a novel sensitivity analysis that can *exactly* characterize the set of states for which a unique LP solution is optimal. Furthermore, we identify two properties of the underlying LPs that affect reusability. Finally, we introduce an algorithm that optimizes LP solutions to generalize well to other states. Our new algorithms significantly reduce the number of necessary LP computations.

## Introduction

The objective of optimal classical planning is to find a cheapest sequence of actions that achieves some goal. One of the main methods for optimal planning is A* search (Hart, Nilsson, and Raphael 1968) with an admissible heuristic (Pearl 1984). Currently, the strongest admissible heuristics are based on *cost partitioning* (Katz and Domshlak 2010). In practice, cost partitions are often optimized using linear programs (LPs).

*Saturated post-hoc optimization* (Pommerening, Röger, and Helmert 2013; Seipp, Keller, and Helmert 2021) is a prominent example for this. It partitions costs according to a weighted sum of *saturated* cost functions. In each state encountered during the search these weights are optimized by an LP which incurs significant computational costs. Höft, Speck, and Seipp (2023) introduce methods to drastically reduce the number of solved LPs without compromising the quality of the heuristic. Their approach uses the concept of *sensitivity analysis* from Operations Research, which determines how changes to an LP affect its optimal solutions. In the context of (saturated) post-hoc optimization, the LPs computing cost partitions for two different states $s$ and $s'$ only differ in the bounds for some constraints. With sensitivity analysis one can often avoid computing a cost partition for $s'$ by cheaply adapting the cost partition for $s$ to $s'$.

In the work by Höft, Speck, and Seipp (2023), sensitivity analysis approximates the set of states where a cost partition can be adapted. Here, we go beyond that and develop a method that can characterize this set exactly.

We show that two properties of the underlying LP influence the relationship between a LP solution and the desired planning heuristic: *degeneracy*, where multiple solutions describe the same heuristic and *non-uniqueness*, where multiple heuristics are optimal. In the first case, improving the constraint formulation of the LP can reduce degeneracy; in the second, improving the columns of the LP can reduce non-uniqueness.

In the presence of multiple equally good solutions, we would like to break ties in favor of solutions that generalize to more states. Therefore, we introduce an algorithm which favors LP solutions that correspond to more versatile cost partitions, increasing reusability. Our empirical evaluation shows that our new methods help to drastically reduce the number of LP computations required for the saturated post-hoc optimization heuristic.

## Background

We consider SAS$^+$ planning tasks (Bäckström and Nebel 1995) with operator costs. The details of planning tasks are not important for this paper, the only relevant part is that a planning task induces a weighted transition system. A weighted *transition system* $\mathcal{T} = \langle S, L, T, cost, s_0, S_* \rangle$ consists of a finite set of *states* $S$, a finite set of *labels* $L$, a finite set of *labeled transitions* $T$: $s \xrightarrow{\ell} s'$ with $s, s' \in S$ and $\ell \in L$, a *cost function cost* : $L \to \mathbb{R} \cup \{-\infty, \infty\}$ that assigns a cost to each label, an *initial state* $s_0 \in S$, and a set of *goal states* $S_* \subseteq S$. We also write $s \in \mathcal{T}$ for $s \in S$. A sequence of transitions leading from a state $s$ to a goal state is called an *s-plan* and an $s_0$-plan is called a *plan*. A plan is *optimal* if it has minimal cumulative cost.

Heuristic search (Bonet and Geffner 2001) is a common way of solving planning tasks optimally. A *heuristic* function $h$: $S \to \mathbb{R} \cup \{-\infty, \infty\}$ estimates the cost of the cheapest $s$-plan for each state $s$. It is *admissible* if it never overestimates the true cost of a cheapest $s$-plan $h^*_{\mathcal{T}}(s)$. Strong representatives of admissible heuristics are *abstraction heuristics* (Edelkamp 2001; Helmert, Haslum, and Hoffmann 2007; Katz and Domshlak 2010; Seipp and Helmert 2018) which simplify the transition system of a planning task with a surjective *abstraction function* $\alpha : S \to S^\alpha$, yielding an *abstract transition system* $\mathcal{T}^\alpha : \langle S^\alpha, L, T^\alpha, cost, \alpha(s_0), S^\alpha_* \rangle$, where $T^\alpha = \{\alpha(s) \xrightarrow{\ell}$

$\alpha(s') \mid s \xrightarrow{\ell} s' \in T\}$ and $S_*^\alpha = \{\alpha(s) \mid s \in S_*\}$.

The preferable way of combining several heuristics admissibly is *cost partitioning* (Katz and Domshlak 2010; Pommerening et al. 2015). A cost partition $C$ in a transition system $\mathcal{T}$ is a tuple of cost functions $C = \langle cost_1, \ldots, cost_n \rangle$, such that $\sum_{i=1}^n cost_i(\ell) \leq cost(\ell)$ for all labels $\ell \in L$. Evaluating a tuple of admissible heuristics $H = \langle h_1, \ldots, h_n \rangle$ for state $s$ under $C$ yields the admissible estimate $h^C(s) = \sum_{i=1}^n h_i(cost_i, s)$, where each heuristic $h_i$ is evaluated under cost function $cost_i$.

A strong heuristic based on cost partitioning is *saturated post-hoc optimization* (SPhO) (Pommerening, Röger, and Helmert 2013; Seipp, Keller, and Helmert 2021), which is defined as follows. Given a transition system $\mathcal{T}$ and a tuple of abstraction heuristics $H$ for $\mathcal{T}$, the heuristic value $h^{\text{SPhO}}(s)$ for a state $s$ is the *objective value* of the following linear program.

SPhO-LP(s): Minimize $\sum_{\ell \in L} \text{cost}(\ell) \cdot Y_\ell$ subject to

$$\sum_{\ell \in L} mscf_h(\ell) \cdot Y_\ell \geq h(s) \qquad \text{for all } h \in H$$

$$Y_\ell \geq 0 \qquad \text{for all } \ell \in L$$

Here, *mscf* is the *minimum saturated cost function* defined as $mscf(\ell) = \sup_{a \xrightarrow{\ell} b \in T}(h_\mathcal{T}^*(a) \ominus h_\mathcal{T}^*(b))$, where the $\ominus$ operator is defined as regular subtraction for finite values. For infinite values, we use $x \ominus y = -\infty$ iff $x = -\infty$ or $y = \infty$, and $x \ominus y = \infty$ iff $x = \infty \neq y$ or $x \neq -\infty = y$.

*Linear programs* (Thie and Keough 2008), such as SPhO-LP, can be written in the canonical form $\max_{x \in \mathbb{R}^n} \{c^\top x \mid Ax \leq b, x \geq 0\}$ where $x$ is the vector of *decision variables*, $c \in \mathbb{R}^n$ is a vector of *objective coefficients*, $A \in \mathbb{R}^{m \times n}$ is the *coefficient matrix*, and $b \in \mathbb{R}^m$ is a vector of *bounds*. To solve a linear program with the simplex algorithm (Bradley, Hax, and Magnanti 1977), $m$ slack variables are introduced to transform all constraints from inequalities to equalities. The simplex algorithm defines solutions by iterating over *bases*. A *basis* is a valid partitioning of variables and slack variables into $m$ *basic* and $n$ *non-basic* variables. We indicate the indices of the basic variables as $\mathcal{B}$ and the indices of the non-basic variables as $\mathcal{N}$. A partitioning into basic and non-basic variables is valid if the matrix $B$ formed by the $m$ columns of $A$ associated with the *basic* variables is invertible. Each basis defines a unique *LP solution* by setting the non-basic decision variables to zero: $x_\mathcal{N} = 0$, and the basic decision variables to $x_\mathcal{B} = B^{-1}b$. The objective value is then obtained as $c_\mathcal{B}^\top x_\mathcal{B}$.

## Exact Sensitivity Analysis

Höft, Speck, and Seipp (2023) establish that performing sensitivity analysis (Gal 1986) on the SPhO-LP can determine whether the computed cost partition for a state $s$ can be cheaply adapted to another state $s'$ without compromising the quality of the heuristic. They introduce four approaches, however, all of these approaches are approximations, so they may fail to recognize that a previously computed cost partition, can be reused for a newly encountered state. The

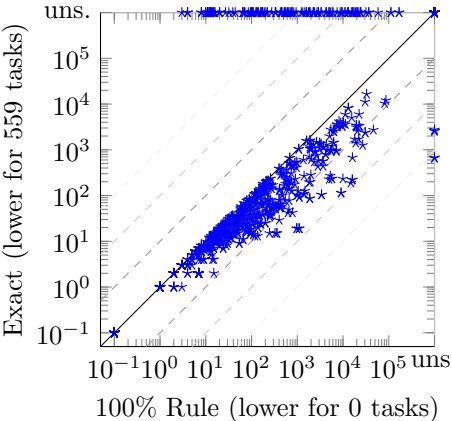

Figure 1: Number of LP solver calls for the SPhO heuristic with exact and 100%-rule-based sensitivity analysis over PDBs for systematic patterns of size 1 and 2.

strongest among them is called "100% rule" and is guaranteed to require at most as many LP computations as any of the other methods (Note that despite its name, the 100% rule also is an approximation.)

A natural question is whether one can go beyond an approximate sensitivity analysis, and if so, to what extent it reduces the number of LP optimizations. To perform such an exact sensitivity analysis, we build on the established result that a basis remains optimal under changes in its constraint bounds $\Delta b$ if and only if all components in the basis header $x_\mathcal{B}^*$ remain greater than or equal to zero i.e, $x_\mathcal{B}^* + \Delta x_\mathcal{B}^* \geq 0$ (Bazaraa, Jarvis, and Sherali 2009). The effects on the basis header from changing the constraint bounds $b$ by $\Delta b$ are captured by the columns of the solution tableau matrix $B^{-1}$, as $\Delta x_\mathcal{B} = B^{-1}\Delta b$ (Vanderbei 2020). Therefore, it holds that $x_\mathcal{B} + \Delta x_\mathcal{B} = x_\mathcal{B} + B^{-1}\Delta b$. And thus, changing the bounds $b$ of an optimally solved LP by $\Delta b$ will preserve optimality of the current solution as long as $x_\mathcal{B} + B^{-1}\Delta b \geq 0$.

Based on this result, we define the following algorithm that maintains a set of basis headers and solution tableau matrices $\langle x, B^{-1} \rangle$: When we encounter a state $s$ during the search, we check whether one of the stored entries satisfies $x_\mathcal{B} + B^{-1}\Delta b \geq 0$. If so, we can use it to efficiently compute the heuristic value $h^{\text{SPhO}}(s)$. Otherwise, we solve SPhO-LP($s$) and add its basis header and solution tableau matrix to our collection. We refer to Höft, Speck, and Seipp (2023) for the algorithm details, as our sensitivity analysis can be used in the same setting. We call this approach the *exact sensitivity analysis* for the $h^{\text{SPhO}}$ heuristic, since it allows us to compute $h^{\text{SPhO}}$ while reusing a previous basis, for *exactly* those states $s$ where this is possible.

Figure 1 compares the number of LP solver calls required to compute the $h^{\text{SPhO}}$ heuristic by exact sensitivity analysis and the 100% rule. This comparison is based on systematic pattern databases up to size 2 (Pommerening, Röger, and Helmert 2013).[1] Overall, we observe a significant increase

---

[1]Höft, Speck, and Seipp (2023) group abstractions with the same minimum saturated cost functions in a single constraint while

in the reusability of cost partitions when using exact sensitivity analysis. It increases the average percentage of evaluations that do not require re-optimizing the LP from 79% to 85%. However, this advantage comes at the cost of slower evaluation and higher memory consumption, which is why some tasks can only be solved using the 100% rule.

## Alternative LP Solutions

All sensitivity analysis methods evaluate the reusability of a basis, not the reusability of the solution defined by the basis. There is a difference between these two cases, because multiple bases can define the same solution coefficients $x_\mathcal{B}$. This means that there can be multiple LP solutions that yield the same cost partition. A necessary condition for this case is *degeneracy*. An LP solution is degenerate if the basis defining it has basic variables of value zero (Bazaraa, Jarvis, and Sherali 2009).

**Proposition 1.** *If an LP solution is degenerate there can be alternative solutions with the same solution coefficients.*

There is also the opposite case of alternative solutions that describe a *different* cost partition, which is captured by the notion of LP solution *uniqueness* (Bazaraa, Jarvis, and Sherali 2009).

**Proposition 2.** *If an LP solution is non-unique there can be alternative solutions with different solution coefficients.*

So far, the topic of alternative optimal solutions for LP-based heuristics has not been discussed in the planning literature, because it is irrelevant when optimizing the LP for every state. However, when reusing LP solutions between different states, alternative solutions are of high interest, as their reusability can be very different. With the definitions of degeneracy and uniqueness, we can establish a precise relationship between sensitivity analysis and cost partitioning.

**Proposition 3.** *Given an LP solution sol for SPhO-LP(s) inducing cost partition $C$, sensitivity analysis can determine the exact set of states for which $C$ is optimal iff sol is non-degenerate and unique.*

*Proof.* Since *sol* is non-degenerate and unique, there are no other solutions for SPhO-LP($s$). Therefore, there are also no other cost partitions besides $C$ for $s$ that yield $h^{\mathrm{SPhO}}(s)$. As a consequence, *sol* is reusable for exactly those states $s'$ for which $C$ is optimal. $\square$

We analyze the extent of degeneracy and uniqueness on SPhO LP solutions in Figure 2. It shows that the LPs computed for $h^{\mathrm{SPhO}}$ over different sets of pattern database (PDB) heuristics (Edelkamp 2001) have many alternative solutions. The number of degenerate solutions is highest for systematic patterns of sizes 1 and 2 (Pommerening, Röger, and Helmert 2013). Removing patterns of size 1 from the collection decreases the average number of degenerate solutions slightly. A possible explanation for this is that the size-1 patterns are subsets of the size-2 patterns, which can lead to redundant constraints. Considering only size-1 patterns has much

---

Figure 1 uses one constraint per abstraction. We analyze abstraction grouping below.

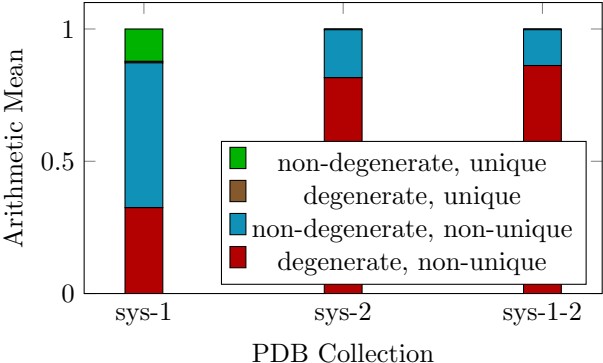

Figure 2: Average percentage of degenerate and non-unique solutions for the SPhO heuristic over different PDB sets.

fewer degenerate solutions than the other two variants and even gives rise to some non-degenerate unique solutions.

## Degeneracy and Non-Uniqueness

While a non-degenerate and unique LP solution describes a single optimal basis, degenerate or non-unique LP solutions imply that there are multiple optimal bases. The existence of multiple optimal bases can affect the reusability of a computed LP solution, since sensitivity analysis is defined for a specific optimal basis.

**Example 1.** *Consider three states: $s_0$, $s_1$, and $s_2$. Further, assume that the SPhO LP for state $s_0$ yields two optimal bases, $x_1$ and $x_2$, and performing a sensitivity analysis on $x_i$ allows us to efficiently compute the heuristic value for state $s_i$. LP solvers can provide only one of these optimal bases at a time. So regardless of which basis ($x_1$ or $x_2$) the solver returns, an additional LP computation becomes necessary to derive the heuristic values for both states $s_2$ and $s_3$. However, if we can reformulate the LP in a way that does not affect the computed heuristic value and at the same time reduces the space of optimal bases, ideally resulting in a single optimal basis (non-degeneracy and uniqueness), it becomes more likely that we can avoid the need for such additional LP computations.*

It is therefore interesting to study whether we can reformulate the SPhO-LP to reduce the number of optimal bases by reducing degeneracy or non-uniqueness. Although it is generally not easy to predict whether a given LP will encounter degenerate solutions, there are two known criteria. Duplicated columns can lead to degenerate solutions and duplicated rows can lead to non-unique solutions (Bazaraa, Jarvis, and Sherali 2009).

For the SPhO heuristic, this means that abstractions with the same minimum saturated cost function can affect degeneracy, and labels that have the same minimum saturated cost under all abstractions can affect non-uniqueness. Both observations are not new. Pommerening, Röger, and Helmert (2013) proposed to group duplicate labels, while Höft, Speck, and Seipp (2023) introduced abstraction grouping. Our analysis of optimal bases gives a novel explanation of why these techniques can reduce the number of required

**Algorithm 1** Greedily increase heuristic weights.

1: **procedure** INCREASEWEIGHTS($\mathcal{H}$, $rem$, $s$)
2:     **for** $h \in \mathcal{H}$ with $h(s) = 0$, in random order **do**
3:         $\Delta w = \min_{\ell \in L} \left\{ \frac{rem(\ell)}{mscf_h(\ell)} \middle| mscf_h(\ell) > 0 \right\}$
4:         $w_h \mathrel{+}= \Delta w$
5:         **for** $\ell \in L$ **do**
6:             $rem(\ell) \mathrel{-}= mscf_h(\ell) \cdot \Delta w$

|           | Base | W+  | GL  | GH  | GHL | GHL-W+ |
|-----------|------|-----|-----|-----|-----|--------|
| Base      | –    | 45  | 89  | 58  | 83  | 87     |
| W+        | **81** | –  | 89  | 44  | 76  | 63     |
| GL        | **116** | **102** | – | 85 | 28  | 40     |
| GH        | **233** | **215** | **236** | – | 60 | 63   |
| GHL       | **261** | **258** | **199** | **76** | – | 13   |
| GHL-W+    | **276** | **256** | **222** | **96** | **34** | – |
| Coverage  | 682  | **832** | 787 | 832 | 790 | **832** |

Table 1: Top: Per-task comparison of the number of LP solver calls by exact sensitivity analysis with different extensions. Cell $(x, y)$ holds the number of tasks for which algorithm $x$ needs fewer LP solver calls than algorithm $y$. The extensions are INCREASEWEIGHTS (W+) and grouping labels (GL), heuristics (GH), or both (GHL). Bottom: Number of solved tasks by the different methods.

LP computations. Table 1 also shows empirical evidence that abstraction and label grouping is beneficial.

## Finding Versatile Cost Partitions

As we discussed above, there can be multiple cost partitions $C$ that yield the same heuristic value $h^C(s) = h^{\text{SPhO}}(s)$ for a given state $s$. Therefore, instead of accepting the arbitrary cost partition $C$ that the LP solver finds for state $s$, we hypothesize that it is beneficial to optimize $C$ to obtain a more *versatile* cost partition $C'$. Such an optimized cost partition needs to preserve the estimate for $s$, i.e., $h^{C'}(s) = h^C(s)$, but apart from this requirement we can change it to make it optimal for more other states $s'$ than the unoptimized $C$.

To obtain more versatile cost partitions, we turn to the dual of the SPhO-LP (Seipp, Keller, and Helmert 2021):

Dual SPhO-LP(s): Maximize $\sum_{h \in \mathcal{H}} h(s) \cdot w_h$ s.t.

$$\sum_{h \in \mathcal{H}} mscf_h(\ell) \cdot w_h \leq cost(\ell) \text{ for all } \ell \in L$$

$$w_h \geq 0 \text{ for all } h \in \mathcal{H}.$$

Intuitively, this LP maximizes a weight $w_h$ for each abstraction heuristic $h \in \mathcal{H}$. In the resulting cost partition $C$, each heuristic $h \in \mathcal{H}$ is assigned the cost function $cost_h$, where $cost_h(\ell) = mscf_h(\ell) \cdot w_h$. The value $h^{\text{SPhO}}(s)$ can then be computed as $h^{\text{SPhO}}(s) = h^C(s) = \sum_{h \in \mathcal{H}} w_h \cdot h(s)$.

When inspecting the SPhO-LP in dual form, it becomes apparent that the LP solver only optimizes the weights $w_h$ for heuristics with a non-zero estimate $h(s)$. All heuristics $h$ with $h(s) = 0$ do not factor into the optimization and their weights can be set arbitrarily by the LP solver. Therefore, we can increase the versatility of a cost partition $C$ by increasing the weights $w_h$ for heuristics $h$ with $h(s) = 0$ to obtain cost partition $C'$. This will preserve the estimate for $s$, but possibly increase the estimates for other states $s'$, making it more likely that $C'$ is optimal for $s'$ than $C$. Additionally, increasing the weights can make the sensitivity analysis more versatile. This is the case, since the new basis can only require fewer changes to be adapted for a new state $s'$.

We define a greedy procedure for increasing weights in Algorithm 1. It expects the remaining costs for all labels $\ell \in L$, which we compute as $rem(\ell) = cost(\ell) - \sum_{h \in \mathcal{H}} mscf_h(\ell) \cdot w_h$. Then the procedure iterates over the heuristics $h$ with $h(s) = 0$ in a random order and increases the weight $w_h$ as long as the remaining label costs allow for an increase.

If the INCREASEWEIGHTS procedure found a higher weight for at least one heuristic, we want to feed this solution back into the LP solver, so that its sensitivity analysis can benefit from the change. We do so by solving the same SPhO-LP again, but warm-starting it with the optimized weights. Re-solving the LP is cheap as long as the LP solver detects that the provided solution is already optimal.

If negative costs occur in the minimum saturated cost functions and the provided solution is degenerate or non-unique, the simplex algorithm may fail to detect its optimality, reject it and searches its own solution again (Bazaraa, Jarvis, and Sherali 2009). In our experimental analysis of SPhO with INCREASEWEIGHTS, we therefore skip the procedure when we detect negative saturated costs. This happens for roughly 30% of our benchmark tasks.

The number of times the INCREASEWEIGHTS procedure finds a weight to increase differs a lot between planning domains and even between tasks from the same domain. In the Petri-Net-Alignment domain, for example, weights are increased between 0% and 21% of the time after finding an LP solution. As Table 1 shows, using INCREASEWEIGHTS usually decreases the number of LP solver calls. It is also raises the number of solved tasks from 682 to 832 for plain exact sensitivity analysis and from 790 to 832 if we group heuristics and labels.

## Conclusions

This paper focuses on the reusability of LP solutions computed for determining heuristic values for saturated post-hoc optimization. We introduced an exact sensitivity analysis approach that improves over existing approximations and empirically showed that it drastically reduces the required LP solver calls compared to the previous state of the art. Furthermore, we showed the importance of considering multiple alternative LP solutions and their potential to generalize to other states. Based on this insight, we proposed a novel greedy approach to optimize cost partitions for increased versatility, resulting in an even greater reduction in the number of LP computations required.

An interesting avenue for future work is to explore methods that not only aim to increase the versatility of a given LP solution, but also directly optimize for its reusability.

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
