# OpenReview forum: "Versatile Cost Partitioning with Exact Sensitivity Analysis"
_icaps-conference.org/ICAPS/2024/Conference — ICAPS 2024_

### Official Review · Reviewer_yQii · 2024-01-16

**Significance And Importance:** 1
**Soundness:** 3
**Novelty:** 2
**Clarity:** 4
**Overall Evaluation:** 1
**Confidence:** 4

**Weaknesses:**

0: Minor weaknesses requiring some work to be addressed for the paper to be accepted.

**Contributions Of The Paper:**

This paper investigates criteria and strategies for re-using LP solutions when deriving admissible state-space search heuristics based on cost partitioning (via saturated post-hoc optimization, in particular).

The paper builds on recent work where sensitivity analysis is used to determine criteria under which an LP solution (a cost partition) remains optimal, with the aim of reducing the computation cost for evaluating the heuristic.

Whereas the existing work relied on approximate criteria, this paper performs a formal analysis to identify exact conditions for reusing solutions, together with problematic situation (e.g. degeneracy). The paper also includes an empirical evaluation of the proposed approach.

**Ethical Considerations:**

(1) Not Applicable: The paper does not have any ethical considerations to address

**Nomination For Best Paper:**

No

**Questions For Authors:**

- Is it indeed the case that in most situation the exact approach has a higher run-time compared to the approximate one?
- Is my reading of table 1 correct? That would suggest that the W+ symmetry breaking approach is not working particularly well

**Reproducibility:**

3: Authors describe the implementation and domains in sufficient detail.

**Strengths Of The Paper:**

The main idea in the paper is correct and it comes with good motivation: reducing the number of solved LPs might in principle improve the practical applicability of an approach that would otherwise have too strong of an overhead.

The analysis is theoretically sound, making good use of classical results from LP theory.

The paper emphasis is on evaluating the main conjecture about the possibility to reduce the number of solved LPs, and the empirical evaluation is (mostly) well designed when viewed from such perspective.

**Weaknesses Of The Paper:**

The paper is not not devoid of weaknesses. In the first place, the contribution w.r.t. Höft et al. (2023), although meaningful, feels quite incremental: both work focused on reusing LP solutions, with the main difference being the use of exact vs approximate methods.

Although never unequivocally stated, the text contains several comments that suggest that the exact approach is in fact performing worse than the approximate one in terms of run-time. This is hinted by a comment at page 3, by the lack of plots referring to run-times, and by the fact that a new symmetry-breaking technique is introduced (the weight-increase approach), despite apparently working less well than existing methods. These issues do not significantly impact the theoretical value of the investigation, but they reduce its practical significance.

I suspect that alternative approaches to amortize the computation overhead may have chance to work better in this setting. Rather than focusing on reusing optimal LP solutions, the authors may also consider:

- Incremental computation, where earlier LP solutions are repaired rather than reused as they are; this is especially appealing since cost-partitioning is meant to be used during tree search, on states that might change in minor ways from parent to child node. This approach is cornerstone of MILP branch & bound, where it is typically performed via the dual simplex algorithm.

- Reusing LP solutions, even if they are not optimal. Here the idea is that any feasible solution will yield an admissible heuristic. This assumes that LP solutions can stay feasible, or that they can be repaired even suboptimally at a very-low computational cost; in this context, the frequency of solution of LPs could become a configurable parameter that one could adjust to obtain the best run-time.

- Consider primal-dual solutions (e.g. Lagrangian relaxations with multipliers optimized via sub-gradient ascent) as a repair approach. This is somewhat complementary to the previous lines, since this kind of approaches provides optimality guarantees at convergence, but can be stopped before convergence and still yield a feasible solution.

For this specific paper, I suspect that making it clear that the emphasis is on a theoretical evaluation may lead to a more consistent message.

Finally, providing access to the code would improve reproducibility (though the technique is simple enough that could be implemented without too much difficulty)


Minor points:

* pg 2: clarify whether evaluating mscf requires double state enumeration, or whether more efficient procedures can be used
* pg 2: unless I am mistaken, the "basis header" mentioned here refers to the tableau header (a basis is just a set of columns and, rather than a table); a more proper term would be "reduced cost vector", "vector of reduced costs", or something along these lines
* pg 2: storing inverted matrices, i.e. B^{-1}, is typically avoided due to the high risk of numerical instability inherent in the inversion operation; if any technique to avoid inversion is employed in the paper, it would be worth mentioning; if no such technique is employed, then numerical instability is an issue to keep in mind

---

> ### Author Rebuttal · Authors · 2024-01-27
>
> Thank you for the excellent suggestions for future work!
>
> We will make our code and experiment data available upon publication.
>
> > Is it indeed the case that in most situations the exact approach has a higher run-time compared to the approximate one?
>
> Yes, the exact method has a higher runtime than the approximate method in most scenarios, so the benefit of being exact does not translate into solving more tasks. We will make this clearer in the paper.
>
> > Is my reading of table 1 correct? That would suggest that the W+ symmetry breaking approach is not working particularly well
>
> Yes, your interpretation of Table 1 is accurate. While the W+ symmetry breaking approach does show improvement, it is not as substantial as the gains achieved by grouping labels or heuristics.

---

### Official Review · Reviewer_7zPQ · 2024-01-21

**Significance And Importance:** 2
**Soundness:** 4
**Novelty:** 2
**Clarity:** 4
**Overall Evaluation:** 2
**Confidence:** 4

**Weaknesses:**

1: Minor weaknesses that are easily fixable.

**Contributions Of The Paper:**

The paper focuses on cost partitioning via post-hoc optimization. Authors show
how to make the post-hoc optimization less computationally demanding by re-using
previous LP solutions. They use so-called sensitivity analysis to decide which
of the previous LP solutions are suitable for the current state, they propose an
algorithm for finding such LP solutions for a given state, and they provide
empirical evidence that the method, indeed, improves state-of-the-art in the
post-hoc optimization.

**Ethical Considerations:**

(1) Not Applicable: The paper does not have any ethical considerations to address

**Nomination For Best Paper:**

No

**Questions For Authors:**

Do you think a similar approach would somehow be useful also in different
LP-based heuristics such as state-equation or potential heuristics?
(This is just out of curiosity, so do not feel obligated to answer it if you
don't have enogh space or you fell you don't have anything to say on that.)\

==== POST-REBUTTAL ====
Thank you for your answers.

**Reproducibility:**

3: Authors describe the implementation and domains in sufficient detail.

**Strengths Of The Paper:**

The topic is certainly relevant to ICAPS. The paper is interesting, mostly easy
to follow, and it pushes our current knowledge about post-hoc optimization a
little bit further.

I also think there is a chance a similar approach to analyzing linear programs
could be useful in other areas like LP-based heuristics (state-equation,
potential, ...). In particular, potential heuristics are usually computed only
once (or to be precise potential function is computed once) and then used during
the whole search. However, it might be beneficial to re-compute potentials
for certain search sub-spaces. The sensitivity analysis used in this paper might
be a viable option that could help us decide when to re-compute it.

**Weaknesses Of The Paper:**

I do not see any major issue. The proposed method is pretty straightforward and
heavily builds on previous research from OR which I very much appreciate.

There are only few minor easily fixable issues:

The notation B^{-1} is not introduced. I assume it is inverse of matrix B, but
it should be explicitly stated to avoid confusion. Similarly for transposition x^T.

There are also terms related to LP solutions that are not defined or explained,
for example, "basis header" or "solution tableau matrix". It should be if not
formally defined, then at least explained on an intuitive level. Moreover, when
citing a specific results from books, I think it should, in this case, include
also a specific theorem or chapter so that the reader can find it more easily.
After all, ICAPS audience might not be familiar with these results from OR and
it would improve the paper if it also provided a little bit more guidance to
interested readers.

Example 1: I do not see reason to number examples here since it contains exactly
one example. I suggest to include it as an ordinary paragraph starting with "For
example". In this case, I think it actually does not break the flow of the text.

---

> ### Author Rebuttal · Authors · 2024-01-27
>
> Thank you for your comments and suggestions!
>
> In the final version, we will incorporate your suggestions to clarify the presentation.
>
> > Do you think a similar approach would somehow be useful also in different LP-based heuristics such as state-equation or potential heuristics?
>
> We agree that it would be interesting to investigate sensitivity analysis for other heuristics.
>
> Our approach has an interesting connection to potential heuristics, particularly in the context of the PhO heuristic over projections. In PhO, an abstract state in a projection with a goal distance of 10, along with its dual heuristic multiplier of w=0.5, is equivalent to a potential heuristic feature corresponding to the same abstract state with a weight of 0.5. Solving the PhO linear program generates an admissible potential heuristic, and we omit reoptimizing if the heuristic value remains optimal. In cases where re-optimization occurs, a new potential heuristic is computed and added.
>
> In the case of the state-equation heuristic, a similar relation can be used for interpretation. The state-equation heuristic computes an optimal cost partition over atomic projections. If we use atomic projections for (S)PhO, it will yield an optimal cost partition that scales only the (saturated) cost functions. Extending the available cost partitions involves introducing PhO constraints for abstractions using different cost functions. Given enough cost functions, PhO computes an optimal cost partition, making it equal to the state-equation heuristic. Therefore, our approach could be used for the state-equation heuristic or for heuristics between the state-equation heuristic and PhO.

---

### Official Review · Reviewer_LQhf · 2024-01-22

**Significance And Importance:** 1
**Soundness:** 3
**Novelty:** 2
**Clarity:** 4
**Overall Evaluation:** 2
**Confidence:** 4

**Weaknesses:**

1: Minor weaknesses that are easily fixable.

**Contributions Of The Paper:**

The paper mainly concentrates on reusing saturated post-hoc optimization LP (SPhO-LP) solutions when computing the heuristic.

It presents a novel sensitivity analysis method, which reduces the overall number of LP computations. It uses an established result on necessary and sufficient conditions for a simplex basis to remain optimal under changes to the constraint bounds, allowing for a simple check about whether an existing SPhO-LP solution for a state can be used for another state. The proposed method is exact, unlike earlier methods that are approximations for the same reusability problem.

The results indicate that the suggested “exact” method outperforms the prevailing approximate method (100% rule). However, many instances exceeded memory or time due to the overhead associated with sensitivity analysis. The paper does not discuss the exact details of what causes the overhead.

Besides exact sensitivity analysis, the paper examines how the choice of SPhO-LP configuration influences reusability. It studies two features of the LP, uniqueness, and degeneracy, that impact the reusability. The paper discusses methods to lower the degeneracy and non-uniqueness of LP and claims that doing so is more likely to decrease the LP calculations. The paper also proposes a greedy approach that picks weights that enhance the reusability of the LP.

**Ethical Considerations:**

(5) Excellent: The paper comprehensively addresses all of the applicable ethical considerations

**Nomination For Best Paper:**

No

**Questions For Authors:**

What percentage of instances, depicted in Figure 1, remained unsolved with the "exact" method? Also, since we are potentially solving a small number of LPs, what is the reason for slower runs? Is it purely because of exact sensitivity analysis?

As per Table 1, Cell(x, y) holds the number of tasks for which Algorithm x needs fewer LP solver calls than Algorithm y. Also, Cell( Base, W+) = 81 and Cell(W+, Base) = 45. Does this mean Base is more effective in reducing LP solver calls than W+, or am I misreading this table?

I understand that degeneracy and non-uniqueness would affect the reusability of an LP solution. However, I am unclear whether uniqueness and non-degeneracy are more likely to have a positive effect. Is there a theoretical basis for this? Also, it is unclear why increasing weights using the IncreaseWeight method makes the resulting cost partition more likely to be optimal for another state.

**Reproducibility:**

3: Authors describe the implementation and domains in sufficient detail.

**Strengths Of The Paper:**

The paper builds on the work of sensitivity analysis in saturated post-hoc optimization. It introduces a novel method for determining if an SPhO-LP for state s can be reused for s’. This method is exact, unlike the existing approximate methods. The technique is empirically shown to be more efficient in lowering LP computations than the current leading approximate approach (100% rule).

The paper examines how the reusability of the exact method can be enhanced by using a greedy method to configure the LP, which lets it pick cost partitions that are more versatile and reusable.

The paper is well-structured. The contributions are well-explained and straightforward to understand.

**Weaknesses Of The Paper:**

The "exact" method could not solve many problems due to processing overheads. However, the nature of these overheads is unclear. I expect that the smaller number of LP calls would reduce the runtime. Also, the exact count/ percentage of the unsolved instances is not given.

---

> ### Author Rebuttal · Authors · 2024-01-27
>
> Thank you for your comments and questions!
>
> > What percentage of instances, depicted in Figure 1, remained unsolved with the "exact" method? Also, since we are potentially solving a small number of LPs, what is the reason for slower runs? Is it purely because of exact sensitivity analysis?
>
> Exact sensitivity analysis solves 682 tasks while the approximate method solves 818 tasks. We will add these numbers to the paper. The exact sensitivity analysis uses more memory than the approximate method and is slower to run due to its matrix multiplication computations.
>
> > As per Table 1, Cell(x, y) holds the number of tasks for which Algorithm x needs fewer LP solver calls than Algorithm y. Also, Cell( Base, W+) = 81 and Cell(W+, Base) = 45. Does this mean Base is more effective in reducing LP solver calls than W+ [...]?
>
> We agree that the description using $x$ and $y$ can be misleading. In Table 1, the entry in row $r$ and column $c$ indicates the number of instances where algorithm $r$ requires fewer LP calls than algorithm $c$. Thus, W+ requires fewer LP calls than Base in 81 instances, and Base requires fewer LP calls than W+ in 45 instances. We will make this clearer.
>
> > I understand that degeneracy and non-uniqueness would affect the reusability of an LP solution. However, I am unclear whether uniqueness and non-degeneracy are more likely to have a positive effect. Is there a theoretical basis for this? Also, it is unclear why increasing weights using the IncreaseWeight method makes the resulting cost partition more likely to be optimal for another state.
>
> Yes, it is not guaranteed that having only non-degenerate and unique solutions provides better reusability. However, the presented algorithms not only reduce degeneracy but also increase reusability.
>
> Removing duplicate rows and columns reduces the redundancy of an LP by merging two alternative solutions and thus it also combines their sets of reusable states.
>
> Using IncreaseWeights, we transform one solution into another that will always have at least as good, if not better, reusability. This is because increasing the weights of a cost partitioning C can make the new cost partitioning C' (with increased weights) optimal for a newly encountered state for which C was not optimal. This is because C’ remains optimal for the states where C is optimal (by construction) and for other states having higher weights (and thus higher heuristic estimates) makes it more likely that C’ is optimal.

---

### Meta-Review · Area_Chair_XvMb · 2024-02-02

**Recommendation:** Accept (Oral)
**Confidence:** 4

**Metareview:**

The reviewers agree that this is a well and clearly structured paper that makes an interesting and valuable contribution to cost partition.

We ask the authors to follow the reviewers' recommendations and include additional clarifications in the final version of their paper, as well as make the code and experimental data available.

**Ethical Considerations:**

(1) Not Applicable: The paper does not have any ethical considerations to address